# Computational Study of Methane C–H Activation by Main Group and Mixed Main Group–Transition Metal Complexes

**DOI:** 10.3390/molecules25122794

**Published:** 2020-06-17

**Authors:** Carly C. Carter, Thomas R. Cundari

**Affiliations:** Center of Advanced Scientific Computing and Modeling, Department of Chemistry, University of North Texas, Denton, TX 76201, USA; carlycarter@my.unt.edu

**Keywords:** CH activation, methane activation, DFT, divalent, computational, methane

## Abstract

In the present density functional theory (DFT) research, nine different molecules, each with different combinations of A (triel) and E (divalent metal) elements, were reacted to effect methane C–H activation. The compounds modeled herein incorporated the triels A = B, Al, or Ga and the divalent metals E = Be, Mg, or Zn. The results show that changes in the divalent metal have a much bigger impact on the thermodynamics and methane activation barriers than changes in the triels. The activating molecules that contained beryllium were most likely to have the potential for activating methane, as their free energies of reaction and free energy barriers were close to reasonable experimental values (i.e., ΔG close to thermoneutral, ΔG^‡^ ~30 kcal/mol). In contrast, the molecules that contained larger elements such as Zn and Ga had much higher ΔG^‡^. The addition of various substituents to the A–E complexes did not seem to affect thermodynamics but had some effect on the kinetics when substituted closer to the active site.

## 1. Introduction

Carbon–hydrogen bond activation is a useful process in catalysis as light alkanes are abundant and have considerable energy contained within their C–H bonds. Much effort is invested into researching this reaction in order to convert light alkanes, like methane, to higher-value products for industrial uses [1,2]. There are many well-known challenges to C–H activation, however, as these light alkanes prove to be highly inert and thermodynamically stable [3]. Because of this, computational methods can be very useful in being able to quickly identify novel systems that show promise in being able to activate methane, and which may then be worthy of experimental trials.

For the activation of aliphatic C–H bonds, there are several different types of mechanisms via which the reaction may proceed. Conceptually, there are three ways a C–H bond may be cleaved: homolytically (H_3_C^•^/H^•^) or through two heterolytic partitionings, proton (H_3_C^−^/H^+^) and hydride (H_3_C^+^/H^−^) abstraction. The present research was motivated by the potential to achieve hydric methane activation (H_4_C → H_3_C^+^ + H^−^) as compared to a more common deprotonation (H_4_C → H_3_C^−^ + H^+^) mechanism [4]. For methane, proton abstraction is ~100 kcal/mol (gas; ~50 kcal/mol in a polar aprotic solvent) thermodynamically easier than hydride abstraction as determined by high-accuracy ab initio techniques like G3B3. Hence, we sought to identify potential methane activators other than those with the typical electropositive metal–electronegative actor ligand motif [4].

Much research, both experimental and computational, was conducted and it was found that breaking C–H bonds can be achieved using complexes containing transition metals [5,6,7] (with many studies of Earth-abundant metal catalysts having focused on the 3d metal complexes) [8,9,10]. However, some research was done with metal-free catalysts, typically employing frustrated Lewis pair concepts [11,12]. Catalysts containing boron bonded with various other metals including Group 12 elements [13], Be [14], Mg [15], and 3d metals (Cu and Zn) [16] are known in the literature. The aforementioned literature precedents sparked an interest in searching for novel systems that could activate methane. For these purposes, experimentally characterized complexes that contained +2/divalent–+3/trivalent element bonds were considered. Structures reported in the Cambridge Structural Database (CSD) [17] led us to consider compounds for this study that involve bonds between Group 2 (E = alkaline earth metals or the related Group 12, Zn-triad) and Group 13 (A = triels).

For this study, known, structurally characterized model complexes were used as a starting point for the general complexes studied herein (Figure 1). These experimentally known complexes have low coordination numbers at the metal or have typically weakly bound ligands such as tetrahydrofuran (THF), which can be more readily displaced by an alkane substrate. The A–E bond was kept intact, and the substituent on the N was modeled as a methyl group in order that effects of metal/ligand modification be more dominated by electronic than steric factors. The model structure used in this study can be seen in Figure 2.

## 2. Results and Discussion

The reactants modeled in this study to activate methane were inspired by existing complexes that contain an A–E–A bonding motif (A being a triel and E being a divalent metal) [13,14,15,18,19] (Figure 2). The following elements were studied in this research: E = Be, Mg, Zn; A = B, Al, Ga. The optimized A–E bond lengths between the different combinations of A and E can be found in Table 1. In all cases studied, both A–E bonds were identical in length to within 0.01 Å. As noted above, many of the target complexes are two-coordinate with two triel-based ligands coordinated to the central divalent metal. However, some are higher coordination, e.g., the MgB complex, which has two additional THFs ligated to magnesium. We calculated the binding free energy of this solvent molecule (THF) to the MgB complex. It was found that THF is weakly bound to the divalent metal (ΔG_binding_ ~ + 0.7 − 1.2 kcal/mol), suggesting that the energetic expense for methane to displace the solvent molecule on solution would not be too inordinate.

### 2.1. Thermodynamics of 1,2 and 2,1 C–H Methane Addition

When reacting the A–E bond of the studied complexes (Figure 2) with the C–H bond of methane, the reaction may occur with two regiochemistries. The first is 1,2 addition, where the divalent E (Group 2 or 12) binds to the methyl and the triel A binds to the hydrogen. The 2,1 addition is the opposite regiochemistry, whereby the E binds to the hydrogen and the A binds to the methyl group. The visualization of these two reactions can be seen in Figure 3.

The free energies for both methane activation regiochemistries were calculated and compared to see which reaction is more favorable and, thus, is a more likely mechanism from a thermodynamic perspective. In Figure 4, the calculated free energies are plotted for all methane activation reactions for each A/E combination modeled. The formation of 1,2 methane addition products proved to be more thermodynamically favorable than the formation of products of the 2,1 reaction for the beryllium complexes. Interestingly, with the heavier divalent (E = Group 2/12) metals—Mg and Zn—it can be seen that the difference between the calculated free energies of 1,2 and 2,1 methane addition to A–E is very small. The average difference between the 1,2 and 2,1 addition for all nine A/E combinations modeled was −2.2 kcal/mol, indicating that 1,2 addition is typically only slightly lower in free energy than the 2,1 regiochemistry. However, the average difference in ΔG for 1,2 and 2,1 methane C–H addition for the three beryllium molecules was −6.6 kcal/mol. Hence, from a thermodynamic perspective, beryllium compounds more clearly favor 1,2 vs. 2,1 methane addition in comparison to the heavier divalent metals.

Focusing on the impact of triels upon methane activation thermodynamics for both 1,2 and 2,1 addition, an interesting trend can be seen; for a given divalent metal, as the metal changes from boron to aluminum to gallium, ΔG becomes more endergonic in all cases (Figure 4). Therefore, for the complexes studied, from a thermodynamic perspective, boron would be the most favorable for methane activation while gallium would be the least favorable. It can also be seen from Figure 4 that methane activation by the complexes containing boron are exergonic; the calculated Gibbs free energies for both the 1,2 and the 2,1 additions of methane for boron complexes with Be, Mg, and Zn ranged from −8.8 (1,2 addition to the BeB complex) to −1.8 (1,2 addition to the ZnB complex) kcal/mol. The only other exergonic reaction with methane for complexes without boron was with the 1,2 addition to the BeAl compound which had a computed ΔG = −4.1 kcal/mol. In contrast to boron, all the molecules containing the heavier triel gallium were endergonic in their reactions with methane, with ΔG values ranging from 5.7 (1,2 addition to BeGa) to 22.8 (2,1 addition to MgGa) kcal/mol (Figure 4), averaging ΔG ~ 17 ± 7 kcal/mol for the three gallium complexes studied.

### 2.2. Bond Dissociation Free Energies for BeAl Complex

The molecule that yielded the most exergonic reaction (see Figure 4) and lowest free energy barrier (vide infra) for methane activation, the BeAl complex (28.9 kcal/mol), was analyzed further, and the relevant bond dissociation free energies for the reactants and the products for both 1,2 and 2,1 addition were compared. The data from Table 2 show that, for both regiochemistries—1,2 and 2,1 addition—the bond containing Be is much stronger and more exergonic than the bond containing Al for the methyl and hydride ligands. This is consistent with Be complexes being generally more thermodynamically favorable for the activation of light alkane C–H bonds than their heavier divalent counterparts. Furthermore, the data in Table 2 are consistent with the observation that the divalent metals (E) impact the reaction more significantly than do the triels (A). Finally, the data suggest that it is the strength of the Be–H/Me bonds that are formed that determines the favorable thermodynamics of the methane activation reaction. In the next section, we assess if these thermodynamic factors result in a kinetic advantage to beryllium compounds.

### 2.3. Transistion States for Methane Activation

Table 3 lists the calculated free energy barriers of the 1,2 methane activation reactions. The 2,1 addition transition states for methane activation were unstable (optimized to reactants, products, or previously found transition states) and, when constrained in geometry, were much higher in free energy than the isomeric 1,2 transition states. Looking at the trends in ΔG^‡^ for the different triels, gallium complexes had the highest methane activation free energy barriers as compared to their boron and aluminum counterparts, in all cases, except for the BeGa complex, where the boron counterpart (BeB) was higher in energy. The free energy barriers for methane activation by gallium complexes ranged from 33.2 to 58.1 kcal/mol with an average of 46.5 kcal/mol, this value being 2.8 kcal/mol higher than the average for activating complexes that contain boron or aluminum as the triel. Similar to the free energies of reaction, boron was more favorable than the aluminum congener, having a lower methane activation free energy barrier in all cases except for BeAl. Plotting the computed ΔG versus ΔG^‡^ showed no obvious linear correlation (see Appendix A). Among the complexes studied in this research, the BeAl complex had the lowest computed methane C–H activation barrier, ΔG^‡^ = 28.9 kcal/mol, versus separated reactants.

Comparing the trends in ΔG^‡^ (Table 3 and Figure 5) indicates a bigger impact from changing the divalent metal (E) versus changing the triel (A). Figure 5 shows the average free energy barriers for methane activation as a function of the divalent metal for the three different triels (B, Al, Ga). The differences are significant as the calculated average ΔG^‡^ ranged from 32.4 (average ΔG^‡^ of the Be complexes) to 55.9 kcal/mol (average ΔG^‡^ of the Zn complexes) (see blue columns in Figure 5). Methane activation barriers for beryllium-containing complexes were, thus, significantly lower than those computed for the magnesium and zinc complexes by an average of ~ 15–20 kcal/mol. In contrast, as discussed above, the average ΔG^‡^ for changing the triel metal for a given divalent only ranged from 43.4 (average ΔG^‡^ of B complexes) to 46.5 (average ΔG^‡^ of Ga complexes) kcal/mol (see orange columns in Figure 5). Taken together, the present results suggest that beryllium complexes appear to be the most promising in terms of complexes that may activate methane and other light alkanes with strong, aliphatic C–H bonds.

### 2.4. Reaction Coordinates for Methane Activation

The reaction coordinates for the Be complexes are discussed in detail in this section as these had, from a computational perspective, the most attractive barriers and reaction free energies for methane activation. A similar discussion of the results for Mg and Zn complexes is collected in the Appendix A.

#### 2.4.1. BeB Activating Complex

For the activating complex containing Be and B, the geometries of the inorganic reactants and products can be seen in Figure 6. The Gibbs free energies for the 1,2 and 2,1 addition of methane were −8.8 and −5.2 kcal/mol, respectively. From this, it was predicted that the 1,2 reaction is the more thermodynamically favorable process to activate methane than 2,1 addition, as noted above, although both reactions are exergonic.

The free energy barrier for the 1,2 addition of methane to the Be–B bond of the complex was 35.1 kcal/mol. Figure 7 shows the computed free energy diagram for the 1,2 addition of methane to the BeB compound. Table 4 also shows specific bond lengths for all of the Be–A molecules. For the Be–B bond in the ground state (GS), the 1,2 addition TS (transition state), and 1,2 addition products, the bond did not differ significantly. The C–H bond of the methane that broke was 1.45 Å in the TS, which was 0.36 Å (~32%) longer than in its ground state. For the Be–B activating complex, the C–H bond had the lowest percentage difference from GS to TS as compared to Be–Al and Be–Ga. The bond that formed between the methyl C and the Be in the TS was 1.92 Å, which was 0.24 Å (~12%) longer than in the final product. The bond that formed between the H from the methane and the B was 1.64 Å in the transition state, which was 0.45 Å (~28%) longer than this bond in the final product.

#### 2.4.2. BeAl Activating Complex

For the activating complex containing Be and Al, the geometries of the products of methane activation are depicted in Figure 8. The Gibbs free energy for the 1,2 and 2,1 addition of a methane C–H bond to the Be–Al bond of the complex was −4.1 and 3.5 kcal/mol, respectively. Hence, the 1,2 activation regiochemistry was more thermodynamically favorable. The ~7 kcal/mol difference in the 1,2 and 2,1 methane addition free energies is commensurate with those computed for the BeB activating complex discussed in Section 2.4.1. The reactions of the BeAl complex were ~ 4–7 kcal/mol more endergonic than the corresponding reactions for BeB.

The free energy barrier for the 1,2 addition of methane to the BeAl complex was 28.9 kcal/mol, which was the lowest free energy barrier for methane activation among all of the complexes studied in this research (Figure 9). Table 4 also displays specific BeAl bond lengths as they changed from the GS reactants to TS to the product. The Be–Al bond length did not vary as the 1,2 reaction moved from reactants to the TS and from the TS to the products (~2% difference in bond lengths for both cases). The C–H bond of the methane in the TS was 1.50 Å, which was 0.41 Å (~37%) longer than in its GS. The bond that formed between the methyl C and the Be in the TS was 1.87 Å, which was 0.19 Å (~10%) longer than in the final product. The bond that formed between the H and Al was 2.08 Å in the TS, which was 0.51 Å (~25%) longer than the same bond in the final product.

#### 2.4.3. BeGa Activating Complex

For the activating complex containing Be and Ga, the geometries of the inorganic reaction products can be seen in Figure 10. The Gibbs free energies for the 1,2 and 2,1 addition of methane were 5.7 and 14.5 kcal/mol, respectively, which was the greatest difference in computed thermodynamics among the A/E combinations studied in this research. Both methane activation reactions were endergonic, which is consistent with other gallium-containing compounds (see Figure 4).

The energy barrier for the 1,2 addition of methane to the Be–Ga bond of the BeGa complex was 33.2 kcal/mol and, thus, 4.3 kcal/mol higher than the barrier computed for the BeAl congener. Figure 11 shows the computed free energy profile for the 1,2 addition of methane to the BeGa compound. Interestingly, although the reaction was ~10 kcal/mol more endergonic for the BeGa complex versus the BeAl analogue, the increase in ΔG^‡^ was less than half this amount. As noted above, there is no obvious correlation between computed thermodynamics and barriers to methane in the studied complexes. Table 4 also shows specific bond lengths for the Be–Ga molecule. The Be–Ga bond length did not vary as much as the Be–Al complex as the 1,2 reaction moved from reactants to the TS and from the TS to the products (~1% difference in bond lengths for both cases). The C–H bond of the methane that broke was 1.57 Å, which was 0.48 Å (~43%) longer than in its ground state, and ~0.1 Å longer than computed for the BeB and BeAl TSs. The bond that formed between the methyl C and the Be in the TS was 1.84 Å, which was 0.17 Å (~9%) longer than in the final product. The bond that formed between the H from the methane and the Ga was 1.94 Å in the transition state, which was 0.41 Å (~21%) longer than the same bond in the final product.

### 2.5. Calcium–Aluminum Complex

Calcium was also considered for an additional comparison. Given that aluminum as the trivalent showed the most promise for activating methane, the calcium complex was paired with aluminum and compared to the other compounds that contained Group 2 metals with aluminum as the triel (BeAl and MgAl). Table 5 shows the results and provides a comparison among the complexes.

The reaction free energies for both the 1,2 and the 2,1 methane addition seemed to be in line with the larger Group 2 metal used, i.e., the reaction was increasingly more endergonic. Interestingly, the calcium complex’s free energy barrier for methane activation was closer to the BeAl complex, being only 4.8 kcal/mol higher, and significantly lower than the barrier for the MgAl complex, by about 13.5 kcal/mol. As such, calcium complexes with the A–E–A motif would appear to be additional systems worthy of experimental scrutiny.

### 2.6. The Impact of Triel Ring Substituents

The complex that yielded the lowest computed free energy barrier for methane activation, the BeAl complex (ΔG^‡^ = 28.9 kcal/mol), was taken, and various groups (R) of differing electron-withdrawing and -donating abilities were substituted on the triel-containing ring to assess how the free energy barriers changed. The hydrogen that was substituted is highlighted in Figure 12. The R groups modeled in this research were methyl, fluoro, chloro, cyano, hydroxy, and phenyl groups. These groups were introduced to the same site of the complex and their Gibbs free energy and free energy barriers were compared.

From the calculated free energy barriers (Table 6), it can be seen that a single substituent on the carbon backbone had little effect on the reaction’s free energy, as well as the free energy barrier. All of the free energies of reaction for 1,2 addition became more endergonic with every substituent used compared to no substitution (H). In contrast to that, most of the substituents for the 2,1 addition became slightly more exergonic aside from the fluoro (no change) and hydroxy (increased by 1.1 kcal/mol) substituents. Looking at the energy barrier, the substituent making the most difference was the cyano substituent which was lower in energy by ~1.5 kcal/mol than no substituent (H). Cyano proved to marginally be the best substituent as both free energies were exergonic with the lowest energy barrier. However, the present results suggest that the backbone R groups do not have a profound impact on the energy of reaction or on the free energy barrier for the activation of methane to the Be–Al bond of this complex.

### 2.7. Impact of Nitrogen Substitution

Since the changes in substituent on the carbon backbone seemed to have little to no effect, substituents were substituted in place of the methyl groups bonded to the nitrogen of the supporting ligand. To maintain overlap, R groups modeled here were hydrogen, fluoro, chloro, cyano, hydroxy, and phenyl (Table 7). Although slight, it does appear that changes for the substituent on the nitrogen had a bigger impact than substituents on the backbone carbons for energy barriers. Most of the free energies of reaction for the 1,2 addition of methane for the various substituents became more endergonic aside from the hydrogen substituent, which was 0.6 kcal/mol more exergonic. In contrast to that and similar to the carbon backbone substituent trends, most of the substituents for the 2,1 addition became slightly more exergonic aside from hydroxy (no change). As for the energy barriers, two substituents had a major impact; the hydroxy and phenyl groups decreased the energy barrier from no substitution (methyl group) by 3.2 kcal/mol and 6.2 kcal/mol, respectively. The transition state for the cyano group did not converge; thus, it was omitted from Table 7. It does appear that alteration of the substituent on N rather than on the backbone carbon of the supporting ligand had a greater impact on the free energy barriers of methane activation. As such, the latter would appear to be a more profitable avenue for experimental exploration.

### 2.8. 1,3,5-Cycloheptatriene as a Substrate

The BeAl complex was reacted with 1,3,5 cycloheptatriene to compare the free energy change, as well as the free energy barrier differences, in relation to methane. Our initial hypothesis was that there might be more hydridic (less protic) character to activation by these divalent metal–triel complexes versus prototypical transition metal–element active sites. Abstracting a hydride from 1,3,5 cycloheptatriene yields the aromatic tropylium cation. As such, 1,3,5 cycloheptatriene is relatively hydridic for a neutral hydrocarbon, and one might expect it to be a suitable surrogate substrate for experimental studies.

From the calculated free energy of reaction (Table 8), methane activation was about 1.1 kcal/mol more endergonic than activation of 1,3,5 cycloheptatriene. However, the methane substrate had a 1,2-addition barrier that was almost double that of the 1,3,5-cycloheptatriene activation. This shows that 1,3,5-cycloheptatriene is significantly more favorable as a substrate versus methane. The computed barrier was far less than the ca. 30 kcal/mol difference [20] in the homolytic C–H bond strengths of the hydrocarbon. While optimized 1,3,5-cylcoheptatriene and the 1,3,5-cycloheptatrienyl fragment in the TS with BeAl had similar CCCC dihedrals and similar CC bond lengths, the much lower C–H activation barrier for 1,3,5-cycloheptatriene compared to methane implies that perhaps there is some modest hydridic character to the activated H in the transition state.

## 3. Computational Methods

The B3LYP/6-311++G(d,p) [21] level of theory was used to calculate the optimization and frequency of all the given molecules in the gas phase at standard temperature (298.15 K) and pressure (1 atm) using the Gaussian 16 software package [22]. All of the complexes were modeled as neutral with appropriate spin multiplicities; the lowest spin state structure was found, and all energies are reported in kcal/mol. All calculations yielded minima that were defined by having zero imaginary vibrational frequencies, while transition states yielded one imaginary vibrational frequency, which was related to the movement of the hydrogen from the methane substrate to the activating complex of interest. For comparison, the wavefunction method MP2 [23], and the M06 [24] and wB97XD [25] functionals were also used for the most promising system identified (a BeAl complex) in this study with the same basis set employed for B3LYP calculations, 6-311++G(d,p) to consider dispersion corrections. Others reported that the neglect of dispersion corrections could potentially give errors up to 40 kcal/mol of difference for energies [26]. Thus, empirical dispersion corrections were calculated using the GD3 [27] methodology.

However, for the present systems, the various methods tested did not significantly differ from the B3LYP data for both the reaction free energies and the free energy barriers. With this same suite of methods, as well as with B3LYP, solvent effects were also calculated using the SMD formalism (THF was chosen as a prototypical solvent of intermediate polarity) for the most promising complex in this study, a BeAl complex. All comparisons of functionals and calculations employing solvent effect modeling can be found in the Appendix A.

## 4. Conclusions

In the present research, nine different molecules (Figure 2), each with different combinations of A (triel) and E (divalent metal) elements, were reacted with methane to see if the computed ΔG and ΔG^‡^ would be reasonable enough (i.e., ΔG close to thermoneutral, ΔG^‡^ ~30 kcal/mol) to warrant further investigation. The triels (A) modeled herein were B, Al, or Ga, while the E divalent metals were Be, Mg, or Zn.

Among the divalent metals, the activating complexes that contained Be had the lowest calculated free energy barriers for methane activation with a range of ΔG^‡^ = 28.9–35.1 kcal/mol. This suggests that molecules containing beryllium are more likely to activate hydrocarbons than those containing magnesium or zinc, although hydrocarbons whose C*_sp_*_3_–H bonds are more reactive/weaker than methane seem the most plausible substrates. Furthermore, among the divalent metals studied, Be complexes also had the lowest calculated reaction free energies with a range of ΔG = −8.8 to +5.7 kcal/mol for the 1,2-addition reaction with methane. It is further notable that these reactions are not inordinately exergonic, which is often computed for more traditional d-block metal–2p element active sites [28]. Such a thermodynamic situation should facilitate subsequent functionalization and catalyst regeneration steps within a catalytic cycle for alkane functionalization. Thus, taken together, the data indicate that beryllium is the most promising candidate for novel complexes to successfully activate a light alkane such as methane.

The average free energy of 1,2 methane C–H addition was ΔG = 5.9 kcal/mol and the average free energy of the 2,1 addition was ΔG = 8.1 kcal/mol. From these averages it can be seen that the computed difference between the 1,2 addition and 2,1 addition mechanisms is only 2.2 kcal/mol. But Figure 4 indicates that there was a large difference between the two reaction pathways with the molecules that contain Be. The average free energy of the 1,2 addition pathway for molecules that contain Be was ΔG = −2.4 kcal/mol, while the average free energy of the 2,1 methane addition pathway for these same molecules was ΔG = 4.2 kcal/mol, which is a difference of 6.6 kcal/mol, which further shows that beryllium is the most promising divalent metal from among those studied.

Previous DFT studies indicated that, for methane activation through a transition metal catalyst, calculations showed that the system had ΔG of ~ 5–10 kcal/mol and ΔG^‡^ of ~ 20–32 kcal/mol [4,29,30]. Compared to the systems studied herein, the beryllium complexes seem to fit within these ranges (although some systems are exothermic), suggesting that these systems may be experimentally viable.

Adding a substituent to the carbon backbone of the ring (Figure 2) did not affect the free energies and energy barrier significantly. For the substituents studied, the free energies of reaction stayed within 2.6 kcal/mol and within 4.2 kcal/mol compared to the non-substituted 1,2 and 2,1 reaction, respectively. The lack of substituent impact on the carbon backbone in the ring might be due to its placement being two bonds away from the active site, as well as a lack of significant inductive or resonance interactions. Substituents on the nitrogen, however, had a bigger impact on the kinetics than substitutions to the carbon backbone (Table 7) as computed for the hydroxy (ΔG^‡^ = 25.8 kcal/mol) and phenyl (ΔG^‡^ = 22.8 kcal/mol) substituents. These substituents on the nitrogen probably had a bigger effect on the kinetics than on the carbon backbone, likely due to being closer to the active site.

Perhaps more importantly, the present computational results indicate that changes in the thermodynamics and kinetics of 1,2 addition of a methane C–H bond to an A–E bond are more sensitive to changes in the divalent metal (E) than changes to the triel (A). Larger metals in the activating complexes such as Ga and Zn generally had very high ΔG^‡^ and, thus, showed less promise for activating light alkane C–H bonds, as they also were mostly endothermic in their reactions with methane. Additionally, the average free energy barrier for methane activation for the different divalent metals studied ranged from 32.4 to 55.9 kcal/mol (ca. 23.5 kcal/mol range), while the free energy barrier for methane activation ranged from only 43.4 to 46.5 kcal/mol (ca. 3 kcal/mol range) (Figure 5) for the different triels modeled. The range of the energy barrier is much wider for the divalents, suggesting modifications to these metals as being the most profitable avenue for future research.

## Figures and Tables

**Figure 1 molecules-25-02794-f001:**
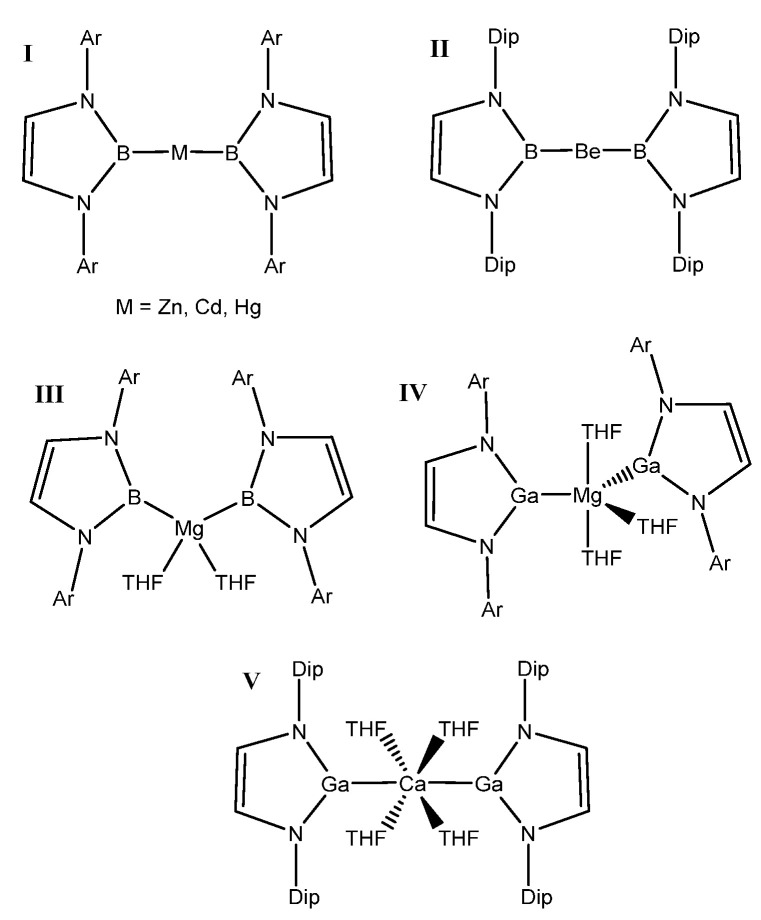
Known compounds with A–E bonds between divalent metals and triels with an A–E–A motif. I [13], II [14], III [15], IV [18], and V [18] (Cambridge Structural Database (CSD) refcodes can be found in the Appendix A).

**Figure 2 molecules-25-02794-f002:**
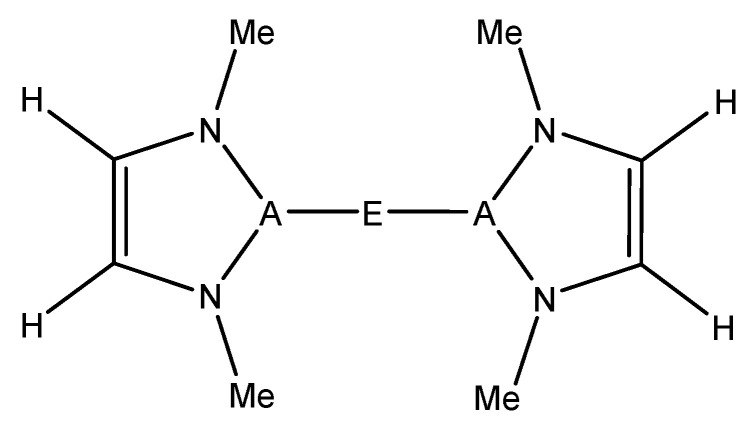
A–E model complexes studied in this research, which were inspired by experimentally characterized complexes. [13,14,15,18,19] E = Be, Mg, Zn; A = B, Al, Ga; Me = methyl.

**Figure 3 molecules-25-02794-f003:**
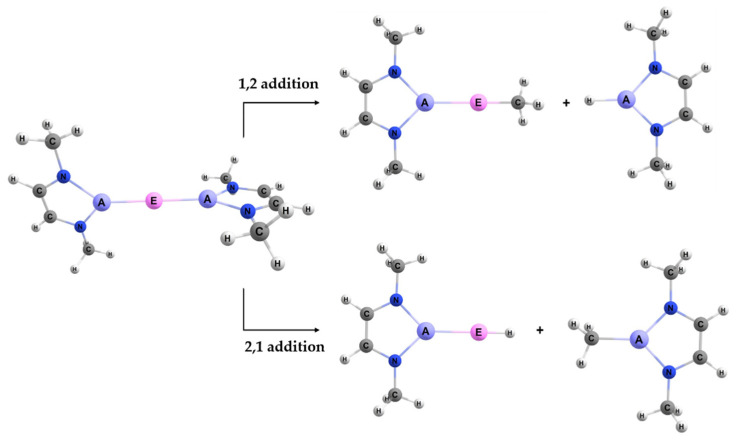
The two regiochemistries studied for activating a methane C–H bond using the A–E bond of the model reactant complexes.

**Figure 4 molecules-25-02794-f004:**
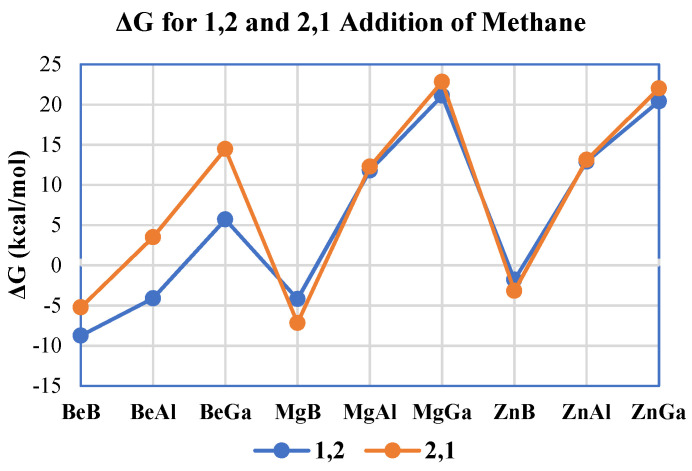
Calculated free energies (in kcal/mol) for both 1,2 (blue) and 2,1 (orange) methane C–H activation regiochemistries for all A–E combinations studied herein.

**Figure 5 molecules-25-02794-f005:**
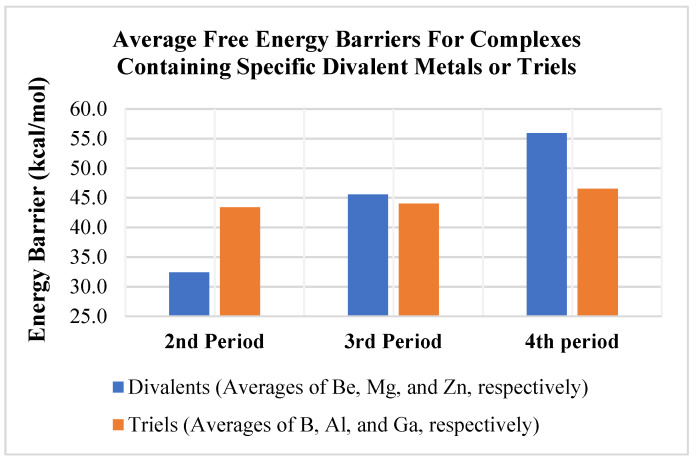
Average calculated free energy barriers for methane activation as a function of divalent metal (blue column) or trivalent element (orange column).

**Figure 6 molecules-25-02794-f006:**
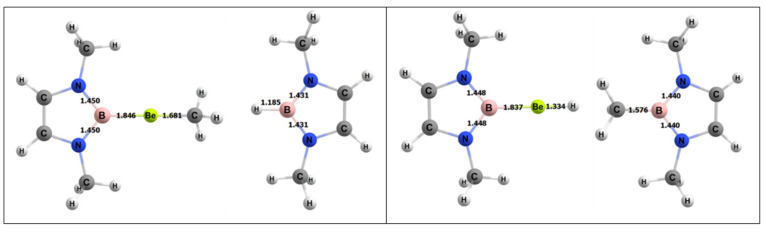
Density functional theory (DFT)-optimized geometries of the products of 1,2 (left) and 2,1 (right) addition of methane to the BeB complex. Bond lengths in Å.

**Figure 7 molecules-25-02794-f007:**
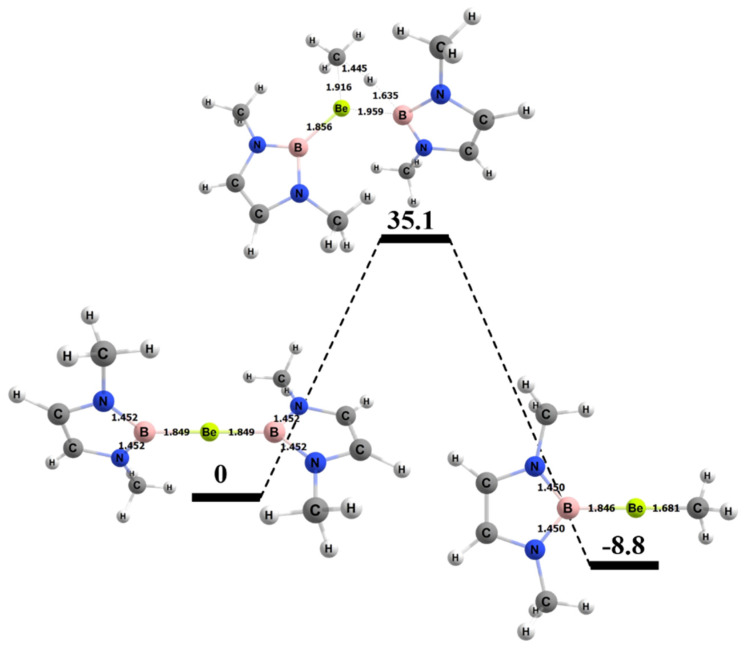
Reaction diagram for the 1,2 addition of methane to the BeB complex. Quoted free energies are in kcal/mol and reported relative to separated reactants (methane plus complex). Bond lengths in Å.

**Figure 8 molecules-25-02794-f008:**
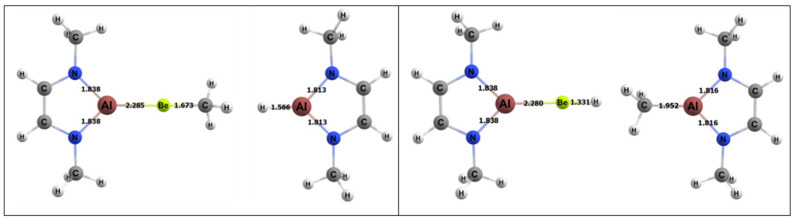
DFT-optimized geometries of the products of 1,2 (left) and 2,1 (right) addition of methane to the BeAl complex. Bond lengths in Å.

**Figure 9 molecules-25-02794-f009:**
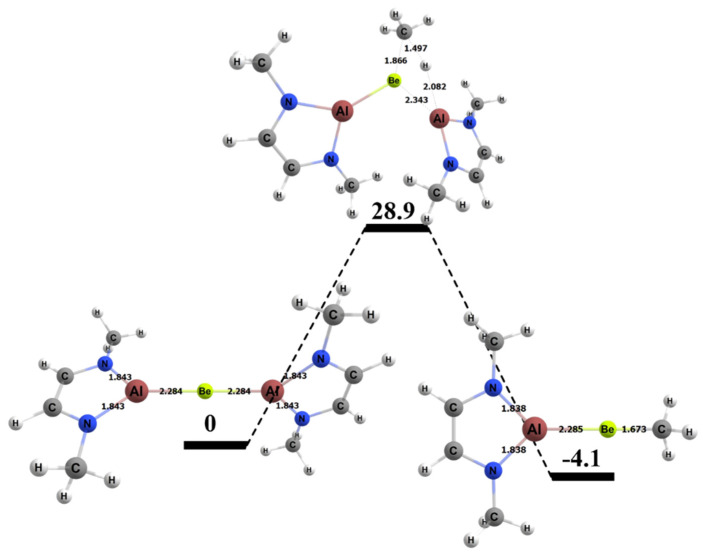
Reaction diagram for the 1,2 addition of methane to the BeAl complex. Quoted free energies are in kcal/mol and reported relative to separated reactants. Bond lengths in Å.

**Figure 10 molecules-25-02794-f010:**
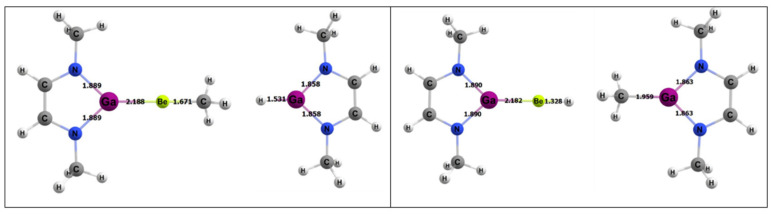
DFT-optimized geometries of the products of 1,2 (left) and 2,1 (right) addition of methane to the BeGa complex. Bond lengths in Å.

**Figure 11 molecules-25-02794-f011:**
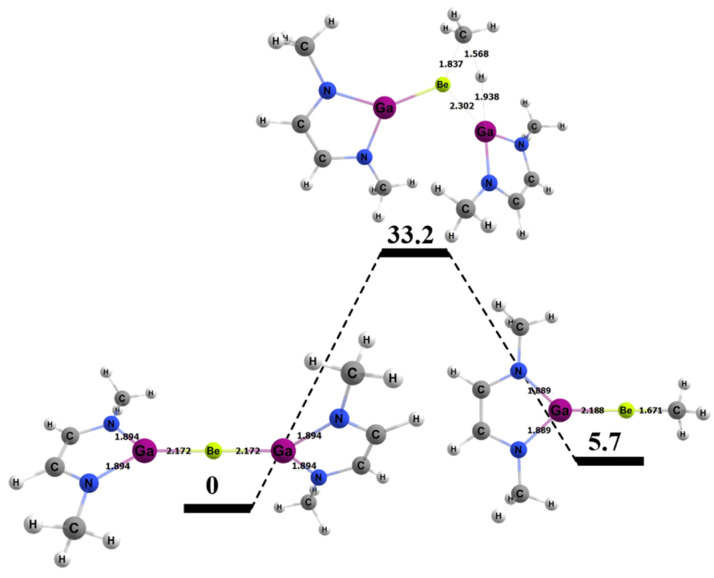
Reaction diagram for the 1,2 addition of methane to the BeGa complex. Quoted free energies are in kcal/mol and reported relative to separated reactants. Bond lengths in Å.

**Figure 12 molecules-25-02794-f012:**
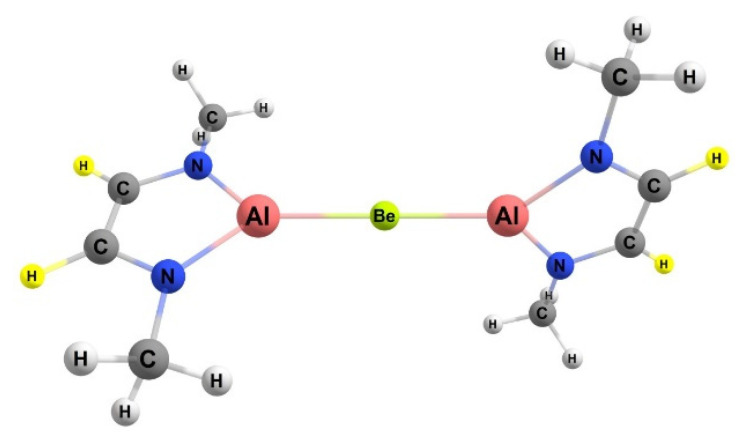
BeAl activating complex and the location of the backbone H (highlighted in yellow) for which R groups (R = methyl, fluoro, chloro, cyano, hydroxy, and phenyl) are substituted.

**Table 1 molecules-25-02794-t001:** B3LYP/6-311++G(d,p) computed A–E bond lengths (Å).

E-A Combination	Optimized Bond Length (Å)
Be–B	1.84
Be–Al	2.28
Be–Ga	2.17
Mg–B	2.26
Mg–Al	2.67
Mg–Ga	2.56
Zn–B	2.07
Zn–Al	2.47
Zn–Ga	2.39

**Table 2 molecules-25-02794-t002:** B3LYP/6-311++G(d,p)-calculated bond dissociation free energies (BDFE) for Be and Al products.

Type of Addition by Methane	Bond of Interest	BDFE (kcal/mol)
1,2	Be–Me	−81.5
	Al–H	−65.6
2,1	Be–H	−84.1
	Al–Me	−55.5

**Table 3 molecules-25-02794-t003:** B3LYP/6-311++G(d,p) calculated free energy barriers of methane 1,2 addition.

E/A Elements	ΔG^‡^ (kcal/mol)
Be–B	35.1
Be–Al	28.9
Be–Ga	33.2
Mg–B	41.2
Mg–Al	47.2
Mg–Ga	48.2
Zn–B	53.8
Zn–Al	55.9
Zn–Ga	58.1

**Table 4 molecules-25-02794-t004:** Bond comparison for all the Be–A complexes. GS—ground state; TS—transition state.

		BeB	BeAl	BeGa
A–Be Bond	GS	1.85	2.28	2.17
	TS	1.85	2.23	2.16
	Product	1.85	2.29	2.19
	TS–GS% Diff	0.2	2.3	0.7
	TS–Product% Diff	0.1	2.3	1.5
C–H Bond	GS	1.09	1.09	1.09
	TS	1.45	1.50	1.57
	%Diff	32.4	37.2	43.7
C–Be Bond	TS	1.92	1.87	1.84
	Product	1.68	1.67	1.67
	%Diff	12.3	10.3	9.0
A–H Bond	TS	1.64	2.08	1.94
	Product	1.19	1.57	1.53
	%Diff	27.5	24.8	21.0

**Table 5 molecules-25-02794-t005:** Comparison of Group 2 complexes (Be, Mg, and Ca) with aluminum.

Complex	1,2 ΔG (kcal/mol)	2,1 ΔG (kcal/mol)	TS 1,2 ΔG^‡^ (kcal/mol)
BeAl	−4.1	−3.5	28.9
MgAl	12.6	11.8	47.2
CaAl	14.9	14.0	33.7

**Table 6 molecules-25-02794-t006:** Calculated Gibbs free energy and free energy barriers for 1,2 addition of methane to BeAl activating complex with various substituents on the triel ring.

Substituent	1,2 ΔG (kcal/mol)	2,1 ΔG (kcal/mol)	TS 1,2 ΔG^‡^ (kcal/mol)
–H (no sub)	−4.1	3.5	29.0
–methyl	−3.7	2.1	29.0
–F	−3.3	3.5	29.3
–Cl	−3.5	2.2	28.3
–CN	−3.6	−0.7	27.6
–OH	−2.0	4.6	29.6
–phenyl	−1.5	2.0	28.3
Average	−3.1	2.5	28.7
SD	1.0	1.7	0.7

**Table 7 molecules-25-02794-t007:** Calculated reaction free energies and free energy barriers for nitrogen substituents on the BeAl complex.

Substituent	1,2 ΔG (kcal/mol)	2,1 ΔG (kcal/mol)	TS 1,2 ΔG^‡^ (kcal/mol)
–methyl (no sub)	−4.1	3.5	29.0
–H	−4.7	3.0	29.5
–F	−2.6	2.3	28.6
–Cl	−2.4	2.7	28.2
–OH	−2.6	3.5	25.8
–phenyl	−2.5	2.7	22.8
Average	−3.1	2.4	27.3
SD	1.0	0.5	2.6

**Table 8 molecules-25-02794-t008:** Comparison of 1,3,5-cycloheptatriene with methane activation reaction free energy and free energy barrier reacted with the BeAl complex for 1,2 addition.

Substrate	ΔG (kcal/mol)	ΔG^‡^ (kcal/mol)
1,3,5-Cycloheptatriene	−5.2	14.8
Methane	−4.1	28.9

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
