# Peer review of "Computational Study of Methane C–H Activation by Main Group and Mixed Main Group–Transition Metal Complexes"

_molecules, 2020, doi:10.3390/molecules25122794_

Round 1
Reviewer 1 Report
Dear Editors and Authors,
Here I provide a brief characterization of the reviewed article:
-The topic of the research is relevant to the hot topic of alkanes functionalization.
-I appreciate the scientific style of the text and the easiness to understand of provided data.
Below I provide a list of questions and suggestions for Manuscript improvement:
1) -> symbol should be replaced by clearer →
2) (H4C -> H3C- + H+) is ~100 kcal/mol (gas; ~50 kcal/mol in a polar aprotic solvent) — phrase is difficult to understand due to multiple brackets
3) ...hydride abstraction (H4 C -> H3C + + H-) as determined by high-accuracy ab initio techniques like G3B3. - if the authors compare two processes it is worth noting both energies and specify here the energy of hydride abstraction.
4) The Discussion section in general lacks the comparison and validation of obtained numerical values or conclusions made on their basis with previously obtained literature data (computational and experimental) for similar tasks. Also, it will be valuable for the readers to understand directly what is the particular advantages of the suggested by the authors type of methane C-H activation with previously developed approaches.
5) In my opinion, as the authors start from CSD search and provide this information on chart 1, the notation of CSD refcodes for the considered compounds can be useful for the readers.
6) In table 1, the authors consider calculated bond lengths. Maybe it will be useful and illustrative to provide and compare the values of angles and torsions in the ground and excited states as it is shown on figs. with energy diagrams.
7) Fig. 3: Addition of vertical notations in the bottom instead of diagonal ones (BeB, BeAl, and so on….) or vertical lines to guide the eye can make the picture more illustrative. Also maybe isolated points but not the lines will be better as it is not a functional dependency but only the trend in different structural rows
8) In my opinion, it is better to put Table 3 after the paragraph of the text but not to start section 2.3 with it.
9) Fig. 4: It is more understandable if one type of data is visualized in a consistent way: either histogram or plot but not both types simultaneously. If the average values are presented maybe the error bars should be present for the more reliable presentation of data.
10) The figures illustrating DFT optimized products (6-13) with different complexes are too numerous and of the same type, maybe some of them should be placed into the Supplementary section or presented in a more compressed form. The main regiochemistry is evident for the reader from the generalized scheme in fig. 2. Also if the authors provide analysis of calculated geometric characteristics, it is better to perform it in a more compressed way so that the comparison of structural differences between different complexes, if they are, becomes more evident.
11) Calcultation section lacks references for the analyzed functionals.
12) Maybe Conclusion section can be slightly shortened so that the provided generalizations are expressed in a more compressed style.
Author Response
We thank the editor and the referees for their hard work, and the helpful nature of their comments. We have carried out additional comments and revised the manuscript in accord with their suggestions to improve the paper. Our responses to their comments are given below in italics.
Here I provide a brief characterization of the reviewed article:
-The topic of the research is relevant to the hot topic of alkanes functionalization.
-I appreciate the scientific style of the text and the easiness to understand of provided data.
We thank the referee for the evaluation of the manuscript, and the positive nature of their comments and helpful suggestions.
Below I provide a list of questions and suggestions for Manuscript improvement:
1) -> symbol should be replaced by clearer →
Thank you for the suggestion, the change has been made in the manuscript.
2) (H4C -> H3C- + H+) is ~100 kcal/mol (gas; ~50 kcal/mol in a polar aprotic solvent) — phrase is difficult to understand due to multiple brackets
Thank you for the suggestion. Some rewording and deletion of redundant brackets have been implemented, plus a short rewriting, to aid with clarity.
3) ...hydride abstraction (H4 C -> H3C + + H-) as determined by high-accuracy ab initio techniques like G3B3. - if the authors compare two processes it is worth noting both energies and specify here the energy of hydride abstraction.
Thank you for the suggestion. We hope that with the rewording of those particular sentences, it will provide clarity.
4) The Discussion section in general lacks the comparison and validation of obtained numerical values or conclusions made on their basis with previously obtained literature data (computational and experimental) for similar tasks. Also, it will be valuable for the readers to understand directly what is the particular advantages of the suggested by the authors type of methane C-H activation with previously developed approaches.
We thank the referee for the suggestion. There is now additional comparison of our computed thermodynamic and kinetic quantities with some found in the literature; Discussion section (4th paragraph, top of page 14).
5) In my opinion, as the authors start from CSD search and provide this information on chart 1, the notation of CSD refcodes for the considered compounds can be useful for the readers.
Thank you for this great suggestion. The CSD refcodes for the listed compounds in Chart 1 are now listed in the SI.
6) In table 1, the authors consider calculated bond lengths. Maybe it will be useful and illustrative to provide and compare the values of angles and torsions in the ground and excited states as it is shown on figs. with energy diagrams.
Thank you for the suggestion. We think that the addition of adding angles and torsions to the figures would make the images appear to be too busy. However, the data can be extracted, if the readers are interested, from the deposited XYZ file.
7) Fig. 3: Addition of vertical notations in the bottom instead of diagonal ones (BeB, BeAl, and so on….) or vertical lines to guide the eye can make the picture more illustrative. Also maybe isolated points but not the lines will be better as it is not a functional dependency but only the trend in different structural rows
We thank the referee for the suggestions in regards to Figure 3. We have made changes that we believe will increase clarity. The labels (BeB, BeAl,… etc.) have been shifted to the bottom of the gridlines and there are now vertical lines to aid in seeing what compound goes to the specific set of data points. Removing the lines was found to be hard to follow for a reader, though we agree it is not a functional dependency, we have decreased the thickness of the lines to still allow the reader to easily read the figure.
8) In my opinion, it is better to put Table 3 after the paragraph of the text but not to start section 2.3 with it.
Thank you for the suggestion. Table 3 has now been moved under the paragraph as it will make more sense to not start a section with a table.
9) Fig. 4: It is more understandable if one type of data is visualized in a consistent way: either histogram or plot but not both types simultaneously. If the average values are presented maybe the error bars should be present for the more reliable presentation of data.
Thank you for the suggestion. Figure 4 now depicts both sets of data in bar graph format for clarity.
10) The figures illustrating DFT optimized products (6-13) with different complexes are too numerous and of the same type, maybe some of them should be placed into the Supplementary section or presented in a more compressed form. The main regiochemistry is evident for the reader from the generalized scheme in fig. 2. Also if the authors provide analysis of calculated geometric characteristics, it is better to perform it in a more compressed way so that the comparison of structural differences between different complexes, if they are, becomes more evident.
Thank you for the suggestion. The images of the products for the 1,2 and 2,1 additions have been combined into a single figure for the corresponding A-E complex to help reduce space taken up. A table now compiles (Table 4) some calculated bond lengths of note so as to be easily compared across the different complexes. The table can be seen split between pages 9 and 10.
11) Calculation section lacks references for the analyzed functionals.
We apologize for the oversight. The references for the functionals B3LYP, MP2, M06 and wB97XD have now been included.
12) Maybe Conclusion section can be slightly shortened so that the provided generalizations are expressed in a more compressed style.
We thank the referee for the suggestion. The conclusion has been shortened as to hit on the main topics from the manuscript.

Reviewer 2 Report
The manuscript entitled “Computational study of methane C-H activation by main group and mixed main group-transition metal complexes” by Carter and Cundari explores computationally the activation of methane by combining strong sigma donor ligands with Be, Mg, or Zn. Small molecule C–H activation is an important area of research, and the idea of using non-transition metal systems to achieve it under mild conditions are highly desirable. I think this study can be very useful, but there are some questions that I have prior to its acceptance, as well as a simple comment.
- The manuscript is clearly written, although some obvious typos still remain – e. Line 61 “Known compounds known”, and Line 428 “corylcopper” – please do a few additional checks to ensure no additional unintended errors.
- The calculations all seem to show two coordinate divalent metals. While there is evidence of this for Be, Mg and Zn will likely have at least a few solvent molecules making up the coordination sphere. I thought I had seen an explanation that these are easily exchanged for alkane, but the calculations imply that all two-to-three are lost, and I’m just not sure if this is true, nor that the solvent corrections performed actually address this concern. As such, I think the authors should either adequately explain how a potentially 5 coordinate complex drops to 2 (or 3 if the argument is a dissociative exchange mechanism for the binding of alkane), or show evidence that the higher energy barriers for larger metals aren’t simply a consequence of higher instability due to increased coordinative unsaturation.
- Related to point 2, did the authors ever run calculations with implicit solvent coordination for the Mg and Zn structures – if so, how do the barriers for these compare to the lower coordinate examples?
Author Response
We thank the editor and the referees for their hard work, and the helpful nature of their comments. We have carried out additional comments and revised the manuscript in accord with their suggestions to improve the paper. Our responses to their comments are given below in italics.
1) The manuscript is clearly written, although some obvious typos still remain – e. Line 61 “Known compounds known”, and Line 428 “corylcopper” – please do a few additional checks to ensure no additional unintended errors.
Thank you for the suggestion. The manuscript has been reread and all typos and errors were fixed.
2) The calculations all seem to show two coordinate divalent metals. While there is evidence of this for Be, Mg and Zn will likely have at least a few solvent molecules making up the coordination sphere. I thought I had seen an explanation that these are easily exchanged for alkane, but the calculations imply that all two-to-three are lost, and I’m just not sure if this is true, nor that the solvent corrections performed actually address this concern. As such, I think the authors should either adequately explain how a potentially 5 coordinate complex drops to 2 (or 3 if the argument is a dissociative exchange mechanism for the binding of alkane), or show evidence that the higher energy barriers for larger metals aren’t simply a consequence of higher instability due to increased coordinative unsaturation.
We thank the referee for this great suggestion. We have now completed a small series of calculations for the complex Mg-B – which is known to have two THF ligated to magnesium - and thus increased its coordination number from 2 to 4 with solvent molecules (THF) to examine the binding free energies of the THF ligands. We found the binding free energy for the addition of one THF molecule to the 2 coordinate system is about +0.7 kcal/mol and the binding energy for a second THF molecule is about +1.2 kcal/mol. So, both mildly endergonic once the entropic factors are incorporated. From these calculations, it is indicated that THF is very weakly bound and displacing it with methane would not be an inordinate energetic penalty. Additionally, we note that most of the divalents chosen in this article (Be and Zn) are already 2-coordinate as well as many other divalent metals with this type of supporting ligand that are found in the CCDC (Cd, Hg, Sn, Pb, etc.).
3) Related to point 2, did the authors ever run calculations with implicit solvent coordination for the Mg and Zn structures – if so, how do the barriers for these compare to the lower coordinate examples?
We thank the referee for raising this point. The impact of implicit solvent effects (THF, e ~ 7) were indeed assessed for the most promising candidate, BeAl (data can be seen in the SI). From the results of this analysis, the free energy barriers changed very little (~ +1 kcal/mol) for the level of theory used throughout the manuscript.

Round 2
Reviewer 2 Report
I am satisfied by the additional experiments/arguments, and recommend publication.